# Janus Kinases in Leukemia

**DOI:** 10.3390/cancers13040800

**Published:** 2021-02-14

**Authors:** Juuli Raivola, Teemu Haikarainen, Bobin George Abraham, Olli Silvennoinen

**Affiliations:** 1Faculty of Medicine and Health Technology, Tampere University, 33014 Tampere, Finland; juuli.raivola@tuni.fi (J.R.); teemu.haikarainen@tuni.fi (T.H.); bobin.george.abraham@tuni.fi (B.G.A.); 2Institute of Biotechnology, Helsinki Institute of Life Science HiLIFE, University of Helsinki, 00014 Helsinki, Finland; 3Fimlab Laboratories, Fimlab, 33520 Tampere, Finland

**Keywords:** leukemia, Janus kinases, kinase inhibitor

## Abstract

**Simple Summary:**

Janus kinase/signal transducers and activators of transcription (JAK/STAT) pathway is a crucial cell signaling pathway that drives the development, differentiation, and function of immune cells and has an important role in blood cell formation. Mutations targeting this pathway can lead to overproduction of these cell types, giving rise to various hematological diseases. This review summarizes pathogenic JAK/STAT activation mechanisms and links known mutations and translocations to different leukemia. In addition, the review discusses the current therapeutic approaches used to inhibit constitutive, cytokine-independent activation of the pathway and the prospects of developing more specific potent JAK inhibitors.

**Abstract:**

Janus kinases (JAKs) transduce signals from dozens of extracellular cytokines and function as critical regulators of cell growth, differentiation, gene expression, and immune responses. Deregulation of JAK/STAT signaling is a central component in several human diseases including various types of leukemia and other malignancies and autoimmune diseases. Different types of leukemia harbor genomic aberrations in all four JAKs (JAK1, JAK2, JAK3, and TYK2), most of which are activating somatic mutations and less frequently translocations resulting in constitutively active JAK fusion proteins. JAKs have become important therapeutic targets and currently, six JAK inhibitors have been approved by the FDA for the treatment of both autoimmune diseases and hematological malignancies. However, the efficacy of the current drugs is not optimal and the full potential of JAK modulators in leukemia is yet to be harnessed. This review discusses the deregulation of JAK-STAT signaling that underlie the pathogenesis of leukemia, i.e., mutations and other mechanisms causing hyperactive cytokine signaling, as well as JAK inhibitors used in clinic and under clinical development.

## 1. Introduction

Cytokine signaling regulates the proliferation, differentiation, and maintenance of cells during hematopoiesis and ensures that a balance of different cell types is maintained in physiological and different stress situations. Most cytokine receptors involved in the regulation of hematopoiesis employ Janus kinase (JAK) for triggering downstream cellular signaling. Ligand binding to cytokine receptors activate the receptor-associated JAKs and lead to phosphorylation and activation of signal transducers and activators of transcription (STAT) (STAT1-6) transcription factors and cytokine-specific gene responses. Apart from hematopoiesis, the JAK-STAT pathway is implicated in several crucial biological processes from embryogenesis to tissue and immune development and inflammatory responses, among others [1].

The JAK-STAT pathway must be tightly regulated to ensure both low tyrosine kinase activity in the absence of cytokine and efficient and transient activation upon extracellular cytokine stimulation. Aberrant regulation of JAK/STAT pathway has been implicated in several autoimmune diseases and malignancies [2]. Currently, ~140 different JAK mutations have been linked to different types of leukemia demonstrating the role of JAKs in leukemogenesis and providing a rationale for their therapeutic targeting.

Currently, six JAK inhibitors have been approved for clinical use, of which two (ruxolitinib and fedratinib) with myeloproliferative neoplasm indication [3,4,5,6,7]. The ongoing development of new JAK inhibitors is aiming for higher compound selectivity and potency with the ultimate goal of disease-specific inhibitors [8]. A deeper understanding of JAK regulation and its roles in immune response and cancers is important for the development of next-generation JAK inhibitors.

### 1.1. Structure and Regulation of Janus Kinases

JAKs are non-receptor tyrosine kinases that constitutively associate with the cytoplasmic region of cytokine receptors and mediate signaling from approximately 60 cytokines and hormones [2]. The four mammalian JAKs (JAK1-JAK3 and TYK2) are ubiquitously expressed, except for JAK3, which is expressed mainly in hematopoietic cells. The JAKs are composed of four domains: N-terminal FERM (4.1 protein, ezrin, radixin, moesin) domain, SH2-like domain, pseudokinase domain (JAK homology-2 [JH2]), and the C-terminal kinase domain (JH1). The FERM and SH2 domains link JAKs with the cytokine receptor juxtamembrane Box1 and Box2 regions. The JH2 domain binds ATP but has low/absent catalytic activity and serves as a key regulatory domain. The JH1 domain is the active tyrosine kinase domain responsible for substrate phosphorylation.

The function and the downstream signaling cascades activated by different JAKs are determined by the characteristics of the interacting cytokine receptor [9,10,11,12]. The cytokine receptors can form either homodimers (identical receptor subunits, e.g., receptors for erythropoietin [EPOR] and growth hormone [GHR], etc.) or heterodimers (different subunits, e.g., receptors for interferon γ [IFNGR1 and IFNGR2] and interleukin 12 [IL12Rβ1, IL12Rβ2]). The heterodimeric receptors harbor two different JAK family members (e.g., JAK2 and TYK2), while the homodimeric receptors bind only JAK2. Binding of a cytokine to the extracellular region of the cytokine receptor initiates signal transduction, leading to receptor rearrangement and/or dimerization, and the associated JAKs undergo conformational changes leading to *trans*-phosphorylation of the kinase domains and stimulation of kinase activity (reviewed in References [13,14]). The exact nature of these structural/conformational changes are not completely understood, but the JH2 domain plays an important role in the activation of the JH1 domain (discussed in the next section). The activated JAK phosphorylates specific tyrosine residues in the receptor chains creating docking sites for SH2 (Src homology 2) domain-containing proteins such as signal transducer and activator of transcription (STAT) proteins [15]. Phosphorylated STATs translocate to the nucleus where they initiate transcription of cytokine responsive genes. JAKs serve as triggering kinases for cytokine signaling and in addition to STATs, they also activate other signaling pathways including MAPK (MEK and P38) and PI3K (Ak1, RS1, and RPS6) pathways which play crucial roles in the proliferation, survival, and differentiation of hematopoietic cells [14,16]. JAK signaling is regulated by a feedback loop where STAT activation leads to the production of negative regulators of JAKs, including SOCS (suppressor of cytokine signaling) and SH2B family of adaptor proteins. SOCS proteins bind to the phosphorylated receptor or JAK, and the suppression of JAK activity occurs through different mechanisms depending on the SOCS in question; physically blocking the STAT binding, ubiquitination, and proteasomal degradation, and through internalization of the receptor-JAK complexes ([17], reviewed in Reference [13]). Negative regulation of JAK-mediated signaling also involves dephosphorylation by protein tyrosine phosphatases, particularly SHP-2, which dephosphorylate activated STATs reducing JAK signaling activity (reviewed in Reference [18]). SHP-2 has also a positive regulatory function in JAK2 signaling pathway: it dephosphorylates JAK2 Y1007, precluding the formation of JAK2-SOCS1 complex, positively contributing to JAK2 stability [19]. JAKs are also subject to regulation through phosphorylation, and in addition to positive regulation through, e.g., phosphorylation of the activation loop tyrosines, negative regulatory sites, such as S523 and Y570 in JAK2, have been identified. Interestingly, the JH2 domain in JAK2 is responsible for S523 and Y570 phosphorylation, and as these sites are not conserved in other JAKs, this regulation appears to be specific for JAK2 and possibly in the homodimeric receptor signaling [20,21].

### 1.2. JAK Regulation by JH2 Domain

The pseudokinase (JH2) domain has been shown to be a critical regulator of JAK activity, having both positive and negative regulatory functions. The positive regulatory role was first identified when loss-of-function mutations in JAK3 JH2 were found to result in severe combined immunodeficiency [22,23]. Likewise, JH2 mutations/domain deletion were found to impair cytokine-induced signaling in TYK2 and JAK2 [24,25,26], supporting the positive regulatory function of the JH2 domain. However, while JH2 domain deletion suppresses cytokine-induced activation, it increases basal signaling activity [25]. Furthermore, many JAK2 mutations in JH2 result in hyperactivation, leading to various malignancies (discussed in the later sections) indicating a negative regulatory role of the JH2 domain. Mutagenesis studies at different regions of the JH2 domain have been shown to suppress the gain-of-function (GOF) disease mutation-induced hyperactivation of JAK, confirming the regulatory role of the JH2 domain [27,28,29,30,31]. A plausible structural basis of the negative regulatory function is that the domain interacts directly with JH1 physically restricting the JH1 *trans*-phosphorylation thus maintaining the domain in an inactive state. Cytokine stimulation or a JH2 domain gain-of-function mutation breaks the autoinhibitory interaction releasing JH1 domain to interact in-trans with the other JH1 domain, facilitating *trans*-phosphorylation and leading to kinase activation [14,32]. The JH2-JH1 autoinhibitory interaction in JAK2 was identified by molecular dynamics simulation and mutagenesis studies, which revealed an interaction between the N-lobes of JH2 and JH1 [33]. A crystal structure of TYK2 JH2-JH1 domains published at the same time revealed an identical autoinhibitory pose [34]. Recent studies using a single molecule co-tracking approach demonstrated a critical role for the JAK2 JH2 domain in thrombopoietin receptor dimerization and stabilization of the ligand-induced dimer [18]. The regulatory role of JH2 makes it an intriguing target for inhibitor discovery [8], and importantly, JAK hyperactivation can be suppressed by mutations inhibiting ATP binding to JH2 [27,30] or targeting the αC helix of JH2 [28,29,31].

### 1.3. JAKs in Hematopoiesis

Cytokines drive the differentiation of hematopoietic stem cells (HSCs) to different types of blood cells in a multi-step process. Signaling during hematopoiesis involves several receptor tyrosine kinases such c-Kit, CSF-1R, and FLT-3 but most hematopoietic cytokines utilize JAK-mediated signaling to regulate survival, maintenance, differentiation, and proliferation of hematopoietic cells. Blood cells have a short life span and need to be renewed constantly. Consequently, hematopoiesis is the most active biological process in our body with 5 × 10^11^ new cells produced every day. The process starts from bone marrow by differentiation of pluripotent HSCs. JAK2 is crucial for the maintenance and function of HSCs as shown in conditional JAK2 knockout mice displaying a rapid loss of HSCs leading to bone marrow failure and lethality [35]. JAK2 is associated with thrombopoietin receptor (TPOR or MPL) and is involved in both early hematopoiesis, as well as in the regulation of megakaryocytes in platelet production [36]. JAK2 is also associated with EPOR, granulocyte/macrophage colony-stimulating factor receptor (GM-CSFR), granulocyte colony-stimulating factor receptor (G-CSFR), IL-3R, and IL-5R and is, therefore, the key signaling mediator in the differentiation of cells in myeloid lineage [12,37,38,39] (Figure 1). Development of the lymphoid lineage is guided by different interleukins and their signaling occurs mainly through the so-called common γc chain receptors for IL-2, IL-4, IL-7, IL-15, and IL-21, where JAK1 associates with the cytokine-specific receptor chain and JAK3 with the γc chain (reviewed by Wang et al. in [40]) [30,41]. The Interferon (IFN) receptors with JAK1/JAK2 in IFN-γ and JAK1/TYK2 combinations in IFN-α/β have their primary function in immune response and inflammation, but they also participate in self-renewal and proliferation of HSCs [42,43,44]. However, the precise role of JAKs associated with IFNα and IFNγ receptors in hematopoiesis and leukemia is yet to be defined.

## 2. JAKs in Leukemia

Considering the crucial role of JAKs in the development and function of hematopoietic cells, it is not surprising that JAK1, JAK2, and JAK3 mutations are often found in leukemia. Although TYK2 is more rarely affected by somatic point mutations, growing evidence suggests that germline mutations in TYK2 can drive leukemia [45]. JAK mutations have been linked to hematological diseases including acute myeloid and lymphocytic leukemia (AML and ALL, respectively), chronic lymphocytic and myeloid leukemia (CLL and CML, respectively), myeloma, and lymphoma. JAK mutations are most frequently found in ALL, which is the most common childhood cancer (25% of cases) [46]. ALL results from the transformation of either immature B or T cells. B-ALL constitutes approximately 85% of ALL cases and has a more favorable prognosis than T-ALL, which has a poor prognosis especially in adults [47]. JAK1 and JAK3 are closely linked to the diseases of lymphoid origin (i.e., ALL), while the aberrant JAK2-signaling affects mainly myeloid lineage. Hyperactivating JAK2 mutations are the key drivers of myeloproliferative neoplasms (MPNs), a group of diseases characterized by abnormal proliferation of hematopoietic progenitor cells in the bone marrow. MPNs include primary myelofibrosis (PMF), polycythemia vera (PV), and essential thrombocythemia (ET) that can progress to AML, as well as the CML, which is characterized by the BCR-ABL1 translocation, the so-called Philadelphia chromosome. PV and ET can also progress to myelofibrosis (MF), or less frequently, myelodysplastic syndrome (MDS) (reviewed in Reference [48]).

This review is focused on JAKs, but other relevant disease-related mutations are also briefly discussed. For example, the altered function of cytokine receptors, STATs, and other proteins (e.g., calreticulin (CALR)) may lead to similar effects as seen with mutated JAKs.

### 2.1. Fusion Proteins

Aberrant JAK activation in human tumorigenesis was originally confirmed by the identification of the ETV6-JAK2 (t(9;12) translocation breakpoint, previously known as TEL-JAK2 fusion protein in ALL and CML patients [49,50]. Other, more rare translocations between JAK2 and PCM1, SSBP2, STRN3, BCR, or PAX5 can also induce lymphoid transformation. These translocations have been detected in patients with BCR-ABL1-like ALL, which is a subtype of the signature BCR-ABL1 translocation in adult ALL (and more rarely in CML). BCR-ABL1-like ALL accounts for 15–30% of B-lineage ALL [51]. A substantial amount of pediatric BCR-ABL1-like ALL cases acquire growth signal independency through constitutive activation of the JAK-STAT pathway [52]. Although translocations in JAK2 have been found in myeloid and lymphoid leukemia, they mostly affect myeloid cells and are often not detectable in a patient’s lymphocyte population [53].

Translocations of TYK2 have been found in lymphoproliferative disorders. In a broad study of Roberts et al., a detailed genomic analysis of 154 patients with BCR-ABL-like ALL was conducted, and ten JAK2 chimeras and a novel TYK2-MYB chimera were detected, with other kinase-activating rearrangements involving, e.g., cytokine receptor like factor 2 (CRLF2), EPOR, and thymic stromal lymphopoietin (TSLP) (these JAK-binding receptors are discussed also in the later chapters) [54]. Cresenzo and colleagues performed a whole transcriptome sequencing for anaplastic lymphoma kinase negative anaplastic large cell lymphoma (ALK-ALCL) samples where they recurrently found a NFkB2-TYK2 hybrid transcript in which the coding region of the NFkB2 is fused to TYK2 [55]. Various cell lines (HEK293T, Jurkat, and mouse embryonic fibroblasts) were used to show that NFkB2-TYK2 is responsible for the constitutive activation of STAT3 in ALCL. In addition, RNA sequencing of the AML cell line MOLM-16 identified a fusion transcript containing the N-terminal region of the RNA binding protein ELAVL like RNA binding protein 1 (ELAVL1) and the catalytic kinase domain of TYK2 [56]. TYK2-NPM1 fusion protein has been identified from cutaneous CD30-positive lymphoproliferative disorder patient samples [57].

### 2.2. JAK Mutations in Leukemia

JAK1 mutations in B-ALL or leukemia of the myeloid origin are rarer than in T-ALL. However, a JAK1 V623A mutation was detected in two AML patients [58], highlighting the ability of constitutively active JAK1 to drive various types of leukemia. Furthermore, Li and colleagues recently identified a JAK1 S646P mutation from four B-ALL patients and showed that the mutant has a high sensitivity to the JAK1/2 inhibitor ruxolitinib [59].

JAK1 V658F, homologous mutation to JAK2 V617F described below, has been identified in adult T- and B-ALL and shown to lead to constitutive JAK1 activation in cell lines [60,61,62]. In addition, the JAK2 disease mutation R683G has a pathogenic counterpart in JAK1 and several mutations at the homologous JAK1 R724 site (namely to His, Gln, or Ser) have been detected in T- and B-ALL cohorts [62].

The first association between JAK2 mutations and abnormal myelopoiesis was obtained from fly studies, where acquired mutations in the Drosophila melanogaster JAK counterpart Hopscotch (Hop) gave rise to a leukemia-like condition [63,64,65]. Mutations in JAK2, its receptor TPOR, CALR, as well as in adapter protein SH2B3 (LNK) (negative regulator of JAK2) are directly linked to Philadelphia negative MPNs [66,67,68]. Majority of MPN patients have a single valine to phenylalanine substitution at residue 617 in the pseudokinase domain of JAK2 [69,70,71]. The JAK2 V617F mutation is detected in more than 95% of patients with PV and in about 50% of patients with ET and PMF. Furthermore, the mutation has been linked to chronic myelomonocytic leukemia, myelodysplastic syndrome, and solid tumors in sporadic cases. Several studies have indicated that crosstalk exists between the JAK-STAT pathway and p53. p53 mutations can co-occur with the JAK2 V617F mutation and drive the progression of MPNs [72]. The majority of MPNs retain wild-type p53, but patients harboring a functional inactivation of p53 have a high prevalence of secondary AML. Acquisition of JAK2 V617F leads to negative regulation of p53 via E3 Ubiquitin-Protein Ligase Mdm2 (MDM2), which degrades p53.

A second mutational hotspot maps to exon 12 into the N-terminal linker region of the pseudokinase domain of JAK2 [14]. These mutants, such as the prevalent K539L mutation, are typically found in V617F negative PV patients, but also in other MPNs. Recently, JAK2 S523L gain-of-function mutation in this region was identified in two MPN patients [73]. This finding also clinically supports the negative regulatory function of S523 phosphorylation in JAK2, as discussed above. The third region with a high mutation frequency in the pseudokinase domain entails exon 16 residues around R683. Within the first decade following the diagnosis, up to 20% of MPN (mainly PMF) patients progress to acute myeloid leukemia (AML), and although de novo JAK mutations in AML are rare, a somatic JAK2 T875N mutation (JH1 interaction site for exon 16 residues) has been detected in about 5.2% of AML patients [74,75].

JAK2 V617F has not been detected in pediatric or adult ALL patients, and in general JAK2 mutations are rare in lymphoid malignancies, especially in comparison to JAK1 or JAK3 mutants [76,77]. However, JAK2 is the most frequently mutated JAK in high-risk B-ALL, although JAK1 mutations and one JAK3 mutation have been found in B-ALL [59,62,78]. Overt JAK2 signaling is also associated with childhood leukemia in Down syndrome (DS) patients. Children with Down syndrome have a 10–20-fold increased incidence of acute leukemia, and JAK2 mutations are found in 20% of DS-ALL [79,80]. The most frequent JAK2 mutations target the same amino acid, R683 that is altered to glycine or serine. This R683G/S mutation is prevalent in both DS and non-DS ALL cases, as well as in B-cell precursor ALL.

Mutations in JAK3 have been found in patients with various leukemia/lymphoma of mostly lymphocytic subtype. JAK3 is mutated in 10% to 16% of T-ALL cases [81], and mutations in the JAK1-binding receptor IL7Rα occur in 10% of the T-ALL patients [82]. JAK1/3 mutations have been associated with early T-cell precursor ALL (ETP-ALL), and JAK3 mutations have been detected in acute megakaryoblastic leukemia [83], extranodal nasal-type natural killer cell lymphoma, and T-cell prolymphocytic leukemia [84]. Similar to JAK1 and JAK2 mutations, the most prevalent mutations in JAK3: M511I [85], A573V, and R657Q can drive several types of leukemia [52,85,86].

TYK2 is associated with the receptors of type I IFN (IFNα/β), interleukin (IL)-6, IL-10, IL-11, IL-12, IL-23, and IL-27. Disease-driving mutations in TYK2 are rarely found in leukemia when compared to other JAK family members [87]. No somatic TYK2 mutations were found in a study with 186 acute adulthood leukemia samples [60] or 424 sporadic ALL cases [88], suggesting that the hematopoietic lineages are more affected by mutations in JAK1, JAK2, and JAK3. However, germline inactivation of TYK2 is associated with auto-immune and immunodeficiency diseases and patients with TYK2 deficiency have bacterial, viral, and fungal infections, highlighting its biological importance [89].

Recently, a set of B-ALL-associated TYK2 variants were found (R425H in FERM domain and I684S, R703W, and R832W in JH2 domain) together with a novel polymorphism S431G [90]. Moreover, the TYK2 expression was diminished in patient bone marrow samples compared to healthy donors, and the study indicated a role for TYK2 in the pathogenesis of B-ALL through the alteration of IFNα signaling and activation of STAT3.

Two germline TYK2 mutations from two pediatric primary acute lymphoblastic leukemia patients were found to activate TYK2, STAT1, 3, and 5. These mutations, P760L and G761V, target the DPG-motive in JH2, and in silico modeling showed that both mutations affect the conformation of the domain [88]. Sanda and colleagues found that the TYK2–STAT1 pathway is activated in T-ALL cell lines [91]. The activation occurred via gain-of-function TYK2 mutations or through activation of IL-10 receptor signaling, leading to upregulation of the anti-apoptotic protein MCL1. In their study, Prutch and colleagues further showed that the IL-10/22-TYK2–STAT1–MCL1 pathway promotes the survival of ALCL cells [92].

In addition, indirect evidence supports the role of TYK2 signaling in leukemogenesis; inactivation of TYK2 in mice was shown to lead to the development of leukemia and lymphoma, plausibly resulting from the impaired immune defense against tumor cells [93]. Furthermore, gene expression profile of the AML cell line MOLM-16 revealed high mRNA levels of JAK2, EPHB4, STYK1, and especially TYK2 in these cells [56].

### 2.3. Pathogenic JAK Activation in Leukemia

Leukemia-associated JAK gain-of-function (GOF) mutations are found throughout the protein sequence and are thus expected to function via a multitude of mechanisms. Several studies have focused on understanding the molecular basis of how these mutations lead to JAK hyperactivation. The majority of activating JAK mutations cluster to the JH2 domain and the JH2-SH2 linker in JAK2, and these sites are mutation hotspots also in JAK1 and JAK3 (Figure 2). Activating mutations have been most thoroughly studied in the context of JAK2 where the molecular and structural basis for both negative (crystal structure and modeling studies) and positive (modeling studies) regulation is understood in some detail [2,14,32,33]. Mutations in the JH2 domain can be grouped according to their receptor dimerization potential [26]. While activating mutations at the SH2-JH2 linker (e.g., M535I and K539L) or near JH2 αC helix (e.g., V617F, H587N, and C618R) induce receptor dimerization, mutations at the JH2-JH1 inhibitory interface (e.g., I682F and R683G) do not. The dimerization-inducing mutations are expected to bypass the need for cytokine to induce receptor dimerization in the JAK activation process. Accordingly, these mutations map to JAK2 JH2 dimer interface in the JH2-JH2 dimer model [26]. Based on structural data mutations mapping at the JH2-JH1 interface, such as JAK2 R683G/S or homologous JAK1 R724H/Q/S and JAK3 R657Q, destabilize the inhibitory JH2-JH1 interface leading to JAK activation. Furthermore, the activation requirements for different JAK2 mutations have been shown to involve distinct mechanisms and JAK domains [28].

While JH2 is the hotspot of JAK mutations, also other domains harbor activating leukemic mutations. In adult T-cell leukemia/lymphoma (ATL), JAK3 FERM domain mutations have been shown to induce gain of function in JAK3 and to activate JAK3 in vitro [94]. FERM domain has been suggested to be involved in the autoinhibition of JAKs [95] and these mutations were suggested to disrupt the autoinhibitory activity of FERM. In addition to FERM domain, JH1 harbors several activating mutations. Majority of these mutations seem to have a similar activation mechanism as non-dimerizing JH2 mutations, i.e., they are located at the inhibitory JH2-JH1 interface and the mutations presumably lead to the disruption of the autoinhibitory interface [33,34,87].

### 2.4. STATs in Leukemia

Altered function of the cytokine receptors or STATs may lead to similar effects as seen with mutated JAKs. Mutations in STAT3 and (to a lesser extent) STAT5 have been evaluated to be frequently leukemogenic in adults with large granular lymphocyte leukemia, and activating somatic STAT3 mutations have been found in T and B cell lymphomas and chronic lymphoproliferative disorders of NK cells [96,97]. STAT5 has a key role in the induction of BCR-ABL-driven ALL and CML [98,99]. Furthermore, mutations in STAT5 are recurrent in T-ALL, and together with the observed mutations in the IL7R and JAK1 and JAK3, they highlight the importance of the IL7R/JAK/STAT5 axis in the T-cell development and function both in health and disease [100].

STATs interact with p53 in both physiological and pathological situations. STAT1 indirectly stabilizes p53, but also directly binds to p53 and coactivates its transcriptional effects on certain apoptotic genes [101]. Activated STAT3 binds the TP53 (the p53 gene) promoter and inhibits the expression of p53 [102]. Co-immunoprecipitation assays have shown an interaction between p53 and STAT5. Interestingly, the interaction was apparently independent of STAT5 tyrosine phosphorylation and p53 inhibited STAT5 transcriptional activity [103]. STAT5 down-regulates the expression of nucleophosmin 1 (NPM1), which is a known stabilizer of p53 [104]. Finally, genes that are induced by STAT5/p53 complex have been found to be overexpressed in MPN patient platelets [103].

### 2.5. Mutations in JAK-Related Receptors and CALR

JAK activation requires interaction and subsequent transphosphorylation of two JAKs which can either occur between two different JAKs in heterodimeric receptors or through two JAK2s in homodimeric receptors. In myeloid cells, JAK2 binds to TPOR/MPL, (EPOR) and G-CSFR and drives STAT3 and STAT5 activation, proliferation, differentiation, and survival of cells. Other homodimeric JAK2 systems include growth hormone (GH) and prolactin (PRL) receptors.

Rearrangements in EPOR are found in ~9% of BCR-ABL-like ALL [105], and TPOR/MPL mutants (W515L/K) cause 5% of ET and PMF subtypes of MPNs [106]. Mutations in the endoplasmic reticulum chaperone protein CALR activate JAK2 signaling via pathogenic interaction with TPOR/MPL, and CALR mutations are causative in 25% of ET and PMF cases [107].

JAK1 associates with IL7Rα, which can form a heterodimeric complex with the γc as well as with cytokine receptor-like factor 2 (CRLF2) in the thymic stromal lymphopoietin (TSLP) receptor. CRLF2 promotes B-cell leukemogenesis and rearrangements in *CRLF2* gene are associated with activating mutations in *JAK1* and *JAK2* and deletions or mutations of the Ikaros transcription factor IKZF1 [108]. The increased expression level of CRLF2 and its interaction with JAK2 correlates with B-ALL that has a BCR-ABL1 ALL-like gene-expression profile and a poor outcome [109].

## 3. Current Status of JAK Inhibitors in MPN/Leukemia

The central role of JAKs in the regulation of immune responses and the identification of somatic mutations in MPNs have spurred the development of JAK inhibitors. Several clinical-stage JAK inhibitors have since been developed through high-throughput screening combined with structure-guided optimization. Currently, two JAK inhibitors have been approved for myeloid malignancies: Ruxolitinib (JAK1/JAK2 inhibitor) for the treatment of primary and secondary MF, and hydroxyurea resistant PV, and fedratinib (JAK2 inhibitor) for PMF. Ruxolitinib was the first JAK inhibitor approved for clinical use against MPNs and treatment with ruxolitinib leads to a reduction in spleen size and symptomatic improvement in patients with MF [110,111]. Ruxolitinib treatment induces some disease-modifying properties, such as delayed bone marrow fibrosis progression [112]. However, molecular remission is unlikely with ruxolitinib monotherapy, as it does not significantly reduce the mutant allele burden.

Fedratinib, a JAK2 selective inhibitor, was recently (2019) approved for the treatment of PMF [4]. Similar to ruxolitinib, fedratinib treatment shows significant symptom improvement and reduction in spleen size. Fedratinib also displayed clinical efficacy in ruxolitinib-resistant or ruxolitinib-intolerant myelofibrosis patients [113]. In addition to ruxolitinib and fedratinib, several other JAK inhibitors are currently in clinical trials against hematological diseases (Table 1).

All approved and clinical stage JAK inhibitors target the ATP-pocket of the kinase domain and bind in the active conformation of the kinase. Compounds with this binding mode are classified as type I kinase inhibitors (Figure 3A,B). However, chronic exposure to type I inhibitors have been found to result in heterodimeric activation of JAK2 by other JAK family members and is seen in cell lines, animal models, and patient samples [122]. This leads to the persistence of MPN cells despite constant JAK2 inhibition. In addition, type I inhibitors have a limited impact on the mutant allele burden, and they increase activation loop phosphorylation of JAK2, which can lead to a life-threatening cytokine-rebound syndrome. More specifically, ruxolitinib has been found to block the dephosphorylation of activation loop Tyr1007/1008 inhibiting ubiquitination and subsequent degradation of JAK2 [123].

Type II kinase inhibitors bind to the ATP-pocket in an inactive conformation, inducing a displacement of the DFG motif, as the sidechain of the phenylalanine of the DFG flips towards the hinge of the active site (Figure 3A,B). CHZ868 and BBT594 are the only type II JAK inhibitors which have been developed, and they target JAK2 [124,125]. CHZ868 has been tested in preclinical MPN and B-ALL models [124,126]. In preclinical MPN models, the inhibitor attenuated myelofibrosis and reduced mutant allele burden. In addition, CHZ868 suppressed the growth of human B-ALL cells and improved survival in B-ALL mice. Importantly, CHZ868 did not induce JAK2 activation loop phosphorylation and was not associated with inhibitor withdrawal signaling. In cell-based studies, CHZ868 treatment did not lead to inhibitor persistence, as observed with type I inhibitors; binding of CHZ868 to JAK2 inhibited transphosphorylation of JAK2 by JAK1/TYK2 thus preventing JAK2 activation. Despite promising preclinical findings, no type II JAK2 inhibitors are currently in clinical trials, possibly due to specificity issues. Type III inhibitors are allosteric, non-ATP competitive compounds. Because they bind outside the highly conserved ATP pocket, they can provide more specific inhibition than compounds targeting the conserved ATP pocket (Figure 3A,B). Two JAK2 inhibitors, ON044580 and LS104, have been found to bind with an ATP non-competitive mode [127,128], although they showed substrate competitive binding modes. Interestingly, ON044580 was only active in vitro in the presence of JH2 domain and the catalytic domain alone was not inhibited by the compound. Precise inhibitory mechanisms and binding sites for these inhibitors have not been determined.

## 4. Prospects for JAK Targeting in MPN/Leukemia

### 4.1. Pseudokinase Targeting

As the majority of activating JAK mutations cluster in the pseudokinase domain, targeting JH2 might provide mutant selective inhibition. This is also indicated by mutagenesis experiments, where point mutations in the JH2 ATP-pocket and surrounding sites show selective inhibition of pathogenic signaling [28]. Several inhibitors have been designed against JAK2 JH2, some of which bind to the domain with high affinity [129,130,131]. However, these compounds have not shown any inhibition of JAK2 kinase activity. On the other hand, compounds targeting TYK2 JH2 have been shown to efficiently inhibit TYK2-mediated signaling and are in phase 3 clinical trials against autoimmune diseases (Figure 3C) [132,133]). As TYK2 JH1 inhibitors have shown activity in T-ALL, TYK2 JH2 inhibitors might have utility also in the treatment of leukemia [134]. The JH2 targeting compounds have been suggested to function through stabilization of JH2-JH1 autoinhibitory interaction. These compounds also bind JAK1 JH2 (albeit with much lower affinity) and have inhibitory activity toward JAK1 signaling. It remains to be seen whether JH2 targeting strategy can be efficiently utilized outside TYK2 signaling and mechanistic differences in signaling through various JAK-receptor complexes may become relevant in the JH2 targeting approaches.

### 4.2. Covalent Inhibitors

Covalent inhibitors lead to a long-lasting inhibition and can provide highly selective target engagement compared to reversible binders. Covalent inhibitors usually target the highly nucleophilic thiol group of cysteine through an electrophilic warhead present in the inhibitor. Although no cysteines have been found to possess a catalytic role in kinase active sites, a cysteine residue is found in ATP-pocket of some kinases. JAK3 is the only member of the JAK family to have a cysteine (Cys909) in the ATP pocket. This residue has been targeted in JAK3 drug discovery, and one highly selective covalent JAK3 inhibitor is currently in phase 3 clinical trials against autoimmune diseases (Figure 3D). While it is not clear whether specific JAK3 inhibition would be beneficial for leukemia therapy, JAK3 T-ALL mutants have been found to be sensitive to inhibition [135].

### 4.3. Combination Therapy

The limitation of type I inhibitors in the treatment of MPNs, including inadequate disease-modifying activity and persistence, has led to the evaluation of combination treatments as therapeutic options. Ruxolitinib has been tested in several clinical trials in combination with other approved drugs. The aims for combining ruxolitinib with other agents can be either increasing the disease-modifying effects or to reduce unwanted side effects, such as treatment-associated anemia.

Consistent with the mechanism of action of JAK2 inhibitors, the main adverse effect of ruxolitinib is anemia, which is observed in ~40% of patients, and thrombocytopenia [136]. The use of erythropoiesis–stimulating agents (ESAs) in combination with ruxolitinib to alleviate anemia was initially discouraged in phase 3 COMFORT I study but some responses were seen in a *post hoc* analysis from COMFORT II patients [137]. In a recent multi-center study, ESAs in combination with ruxolitinib led to improvements in anemia in MF patients, with no associated toxicities and with a trend towards better survival [136]. Ruxolitinib has also been tested in combination with danazol, which has been used for the treatment of anemia in MF patients. Hematologic stabilization was observed with the treatment, which may be clinically relevant. However, no clear conclusions could be drawn from the study partly due to the small study size [138]. Ruxolitinib in combination with low dose thalidomide, stanozolol, and prednisone improved anemia and thrombocytopenia. The main adverse effects induced by the combination treatment were increased ALT/AST ratio and edema but they were manageable [139].

Several trials of ruxolitinb in combination with other disease-modifying agents have been conducted to increase the efficacy of therapy. Azacitidine, a pyrimidine nucleoside analog of cytidine, is a DNA methyltransferase inhibitor used for the treatment of myelodysplastic syndrome and also delaying its conversion to AML [140]. Ruxolitinib and azacytidine were found to display a synergistic effect in a phase 2 study, and the treatment was both safe and effective [141]. Notably, improvements in spleen response rates and in bone marrow fibrosis were observed in the patients. Panobinostat a pan-deacetylase inhibitor, dampens JAK signaling through increased acetylation of the JAK2 chaperone HSP90; hyperacetylation of HSP90 attenuates its functions and leads to polyubiquitylation and degradation of HSP90 client proteins, such as JAK2. Panobinostat as a monotherapy against PMF has shown a reduction in splenomegaly and *JAK2* V617F allele burden in phase 2 studies [142]. Preclinical mouse model studies with ruxolitinib and panobinostat resulted in a markedly improved efficacy compared to monotherapy with either compound, providing a strong rationale for combination therapy [143]. In phase 1/2 studies, treatment with ruxolitinib combined with panobinostat was well-tolerated and resulted in the reduction of splenomegaly [144,145]. Some patients also had reductions in V617F allele burden.

In addition to MPNs, JAK inhibitors have been indicated in the treatment of CML. CML treatment with Bcr-Abl inhibitors targets mature CML cells but does not effectively eradicate leukemia stem cells (LSCs). Accordingly, CML LSCs survive without of Bcr-Abl activity [146,147]. CML cells have increased STAT5 activity and STAT5 signaling is required for CML development in mice [148]. In addition, JAK1-STAT3 signaling is activated in CML in response to BCR-ABL inhibition [149]. Interestingly, JAK1 or JAK2 inhibition in combination with BCR-ABL inhibitors have been shown to deplete CML LSCs in cell lines and mice models [149,150]. This combination approach is also currently undergoing clinical trials (https://www.clinicaltrials.gov/). JAK-STAT signaling can also be targeted in CML via protein phosphatase 2 (PP2A) activation. PP2A is inactivated in CML LSCs, and its reactivation inhibits STAT5 and leads to depletion of the LSCs. Since JAK2 directly phosphorylates PP2A inactivating PP2A [151], combination of JAK2 inhibitors and PP2A activators could be beneficial in CML treatment.

Several other on-going clinical studies are studying different ruxolitinib combinations, e.g., with inhibitors against BET bromodomain [152] and Bcl-2 [153]. So far, it seems clear that for type I JAK2 inhibitors to achieve molecular remission, combination therapy with drugs having synergistic activity is required.

## 5. Conclusions

Janus kinases drive hematopoiesis and immunity, and aberrant JAK activation plays a crucial role in the pathogenesis of leukemia. Several leukemogenic JAK and other mutations have been discovered, which cause JAK activation via a variety of mechanisms. Currently, JAK inhibitors are in clinical trials against leukemia and the first-generation type I JAK inhibitors have been approved for clinical use against MPNs and other indications are being evaluated. However, the current drugs are not curative and new treatment approaches, including combination therapies, are required. Revealing the mechanistic details of physiologic and pathogenic regulation of JAK signaling will aid in the development of next-generation JAK inhibitors with enhanced potency and specificity.

## Figures and Tables

**Figure 1 cancers-13-00800-f001:**
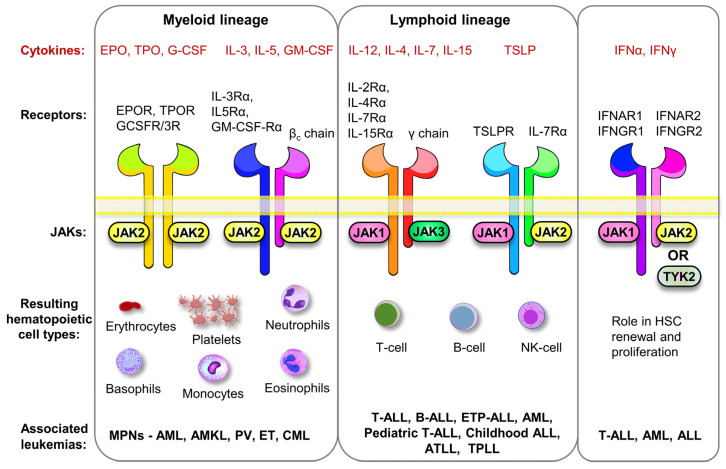
Overview of different cytokines, corresponding receptors, receptor-associated Janus kinases (JAKs), and the resulting hematopoietic cell types formed from the specific JAK/receptor signaling combination. The bottom part of the figure shows the leukemias associated with aberrant JAK regulation in the specific hematopoietic lineage. The interferon (IFN) family does not directly participate in hematopoiesis, although it is crucial for self-renewal and proliferation of hematopoietic stem cells (HSCs), and abnormal JAK regulation in IFN family has been linked to leukemia.

**Figure 2 cancers-13-00800-f002:**
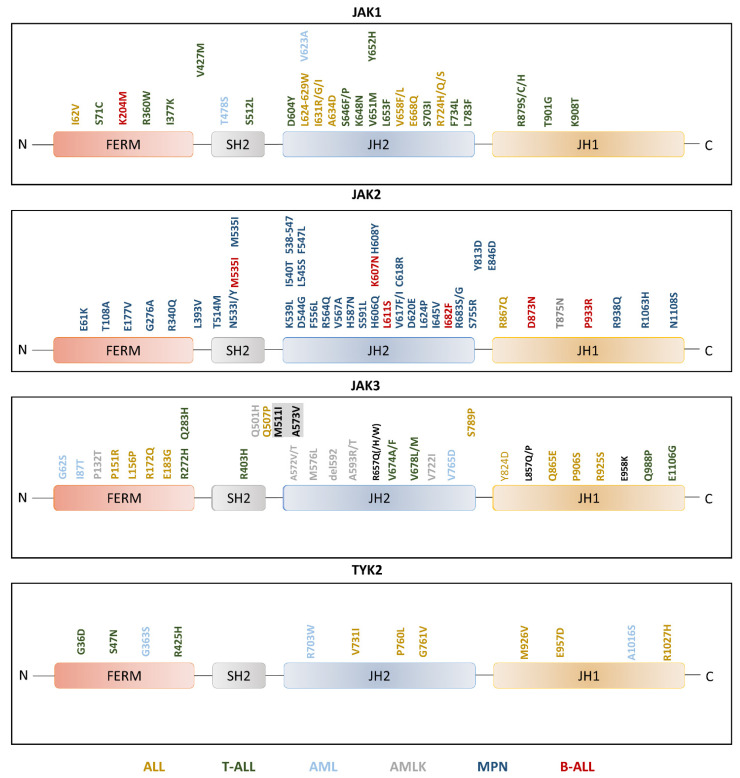
Leukemia-associated JAK mutations and their position in the domain structure. The mutations are colored based on the type of leukemia (acute lymphocytic leukemia (ALL): yellow, T-ALL: green, acute myeloid leukemia (AML): light blue, AMLK: grey, myeloproliferative neoplasms (MPN): blue, B-ALL: red). The figure is based on the table from Hammarén et al. [2].

**Figure 3 cancers-13-00800-f003:**
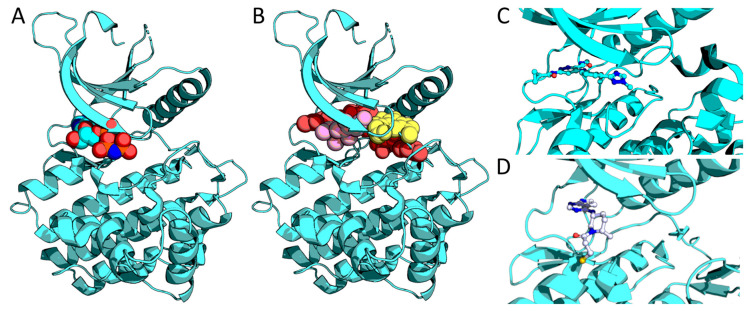
Inhibitor binding sites. (**A**) ATP binding site of JAK kinases. (**B**) Binding sites for type I inhibitors (pink), type II inhibitors (red), and type III inhibitors (yellow). (**C**) Close-up view of a TYK2 JH2-inhibitor complex with a type I binding mode (pdb 6nzp) (**D**) Close-up view of a JAK3 JH1-covalent inhibitor complex (pdb 5toz).

**Table 1 cancers-13-00800-t001:** Clinical stage JAK inhibitors for leukemia/MPNs.

Inhibitor	IC50 (nM)				Condition	Phase	Ref.
	JAK1	JAK2	JAK3	TYK2			
Ruxolitinib	3.3	2.8	428	19	MF, PV	Approved	[114]
					CMML, CLL, SLL, ATL, AML,	Phase 2	
Fedratinib	105	3	1002	405	MF	Approved	[115]
Momelotinib	11	18	155	nd	MF	Phase 3	[116]
Pacritinib	1280	23	520	50	MF	Phase 3	[117]
Lestaurtinib	nd	1	3	nd	AML, MF, PV, ET	Phase 2 *	[118]
Gandotinib	19.8	2.5	48	44	PV, ET, MF	Phase 2 *	[119]
Ilginatinib	33	0.72	39	22	MF	Phase 2 *	[120]
Cerdulatinib	12	6	8	0.5	CLL, SLL, NHS	Phase 2	[121]

* Current status unknown. nd, not determined.

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
