# Peer review of "Janus Kinases in Leukemia"

_cancers, 2021, doi:10.3390/cancers13040800_

Round 1

Reviewer 1 Report

The manuscript is interesting, novel and is well written, clear, and comprehensive. 
The only critical points are as following:

1) the use of many abbreviations, making it difficult to read. I recommend including a list of all abbreviations used in the text and paying attention to write the full names of the acronyms reported in the text.
2) the authors should add a section of Janus  kinases and microRNAs because many miRs targeting Janus Kinase ( for example mir-300, Silvestri G, Trotta R, Stramucci L, Ellis JJ, Harb JG, Neviani P, Wang S, Eisfeld AK, Walker CJ, Zhang B, Srutova K, Gambacorti-Passerini C, Pineda G, Jamieson CHM, Stagno F, Vigneri P, Nteliopoulos G, May PC, Reid AG, Garzon R, Roy DC, Moutuou MM, Guimond M, Hokland P, Deininger MW, Fitzgerald G, Harman C, Dazzi F, Milojkovic D, Apperley JF, Marcucci G, Qi J, Polakova KM, Zou Y, Fan X, Baer MR, Calabretta B, Perrotti D. Persistence of Drug-Resistant Leukemic Stem Cells and Impaired NK Cell Immunity in CML Patients Depend on MIR300 Antiproliferative and PP2A-Activating Functions. Blood Cancer Discov. 2020 Jul;1(1):48-67. doi: 10.1158/0008-5472.BCD-19-0039. PMID: 32974613; PMCID: PMC7510943.)
3) the authors should clarify better the role of Janus kinase inhibitors and quiescent leukemia cancer.
4) the authors should also include the use of PP2A Phosphatase Activator Drugs and if reported, with Janus Kinase Inhibitors together.

Sincerely.

Author Response

We thank the Reviewer 1 for the insightful comments and have considered them carefully. We acknowledge the suggestion (point 2) of adding a section about miRNA. However, due to the lack of comprehensive literature of the subject and the aim for a compact review, we concluded not to add the section, although the subject is interesting. A list of abbreviations has been added at the beginning of the review and unnecessary abbreviations have been removed (point 1). The role of Janus kinase inhibitors and quiescent leukemia cancer, as well as the use of PP2A Phosphatase Activator Drugs is added to the section 4.1. (points 3 and 4). We think these corrections have added value to the review.

Sincerely,

Juuli Raivola

Reviewer 2 Report

SUMMARY

The review describes specific cytokines giving rise to hematopoietic linages shown with corresponding JAK combinations and how dysregulation can lead to leukemia. Select activating JAK mutations are shown across the JAK family with the corresponding leukemic outcome. It is clear that JAKs have become important therapeutic targets and current JAK inhibitors are described along with new selective approaches.

BROAD COMMENTS

The authors clearly stated the aim of the review and explained basic concepts in JAK/STAT signaling with appropriate terminology and scientific abbreviations.

Clear communication of the topic, making it accessible to readers not familiarized with the subject matter. However, general editing is always encouraged.

The figures presented are relevant to the review, however the description of the first figure should be placed in a different section of the text. Similarly, the structure of the text could be slightly revised for better flow. For example, moving section 2.5 “Pathogenic JAK activation in leukemia” within or immediately after 2.2. “Mutations in JAK1, JAK2, JAK3 and TYK2” would make more sense given that mutations and associated leukemias are just covered (see line items).

SPECIFIC COMMENTS

Line 2 capitalize all words in title “Janus Kinases in Leukemia”

Line 24: mention all JAKs without abbreviation (JAK1, JAK2, JAK3 and TYK2)

Lines 26–27 reads better: “JAK inhibitors have been approved by the FDA for treatment of both autoimmune diseases and hematological malignancies.”

Line 38-39 reads better as “Most cytokine receptors in regulation of hematopoiesis employ JAK kinases for triggering downstream cellular signaling. Ligand binding to cytokine receptors activates.”

Lines 48-49 need citation (“Currently approximately 140 different JAK mutations have been linked to different types of leukemia demonstrating the role of JAKs in leukemogenesis and providing rationale for their therapeutic targeting.”)

Lines 50-51 cite the link of the clinical trial and/or reference (“Presently, six JAK inhibitors have been approved for clinical use, of which two (ruxolitinib and fedratinib) with myeloproliferative neoplasm indication.”

Line 70 Remove first call out of Figure 1 and instead keep the call out on Line 137 as the first

Line 107 need citation “However, while JH2 domain deletion suppresses cytokine induced activation, it increases the basal signaling activity.”

Line 126 change “hematopoietic stem cells (HCSs)” for “HSC”

Line 111 put abbreviation in parenthesis Gain of Function (GOF)

Line 131 use superscript for exponential number (5x1011 to 5x1011)

Line 137 need citation for each JAK2 signaling cytokine involved in myeloid linage differentiation “....is thereby the key signaling mediator in differentiation of cells in myeloid lineage.”

Line 140 include Interleukin-21 (IL-21) in the common γc chain receptors

Line 147 fix order of description by describing figure from top to bottom and describe abbreviations

Lines 154-156 terminology will be more accurate as “JAK mutations have been linked to hematological diseases including acute myeloid and lymphocytic leukemia (AML and ALL, respectively), chronic lymphocytic and myeloid leukemia  (CLL  and  CML,  respectively),  myeloma  and  lymphoma.”

Line 164-167 reads better: “MPN entity entails primary myelofibrosis (PMF), polycythemia vera (PV), and essential thrombocythemia (ET) that can progress to AML, as well as the CML which   is   characterized   by   the   BCR-ABL1   translocation, the  so-called Philadelphia chromosome. PV and ET can also progress to myleofobrosis (MF), or less frequently, myelodysplastic syndrome (MDS).” ADD citation as well.

Line 165 substitute thrombocytemia for thrombocythemia.

Line 186 title reads better as “JAK1, JAK2, JAK3 and TYK2 Mutations in Leukemia” and move section 2.5 to 2.2 (as mentioned in broad comments)

Line 212 need citation “A second mutation hotspot maps to exon 12 into the N-terminal linker region of the pseudokinase domain of JAK2.”

Line 226 need citation “Overt JAK2 signaling is also associated with childhood leukemia in Down Syndrome (DS) patients that have 20% higher ALL occurrence.”

Line 255 need citation “Translocations of TYK2 have been found in lymphoproliferative disorders.”

Line 256 need citation "Gene expression profile revealed high mRNA levels of JAK2, EPHB4, STYK1, and TYK2 in AML cell line MOLM-16.”

Line 298 please list citations or create supplementary table with each activating JAK mutation used in Figure 2 

Line 298 interpretation of distribution of mutations for JAK1 is not in agreement with Figure 2. “The majority of activating JAK mutations cluster to the JH2 domain and the JH2-SH2 linker in JAK2 and are prominent also in JAK1 and JAK3, although in JAK1 and JAK3 the mutations are more evenly distributed (Figure 2). The even distribution is seen in JAK3 but not JAK1. In Figure 2 JAK1 has a good cluster within the JH2.

Line 328 correct spelling of  hydroxyurea (one word)

Lines 335-336 cite the link of the clinical trial and/or reference for the approval of Fedratinib.

Line 411 need citation “Consistent with the mechanism of action of JAK2 inhibitors, the main adverse effect of ruxolitinib is anemia, which is observed in ~40% of patients, and thrombocytopenia.”

Author Response

We are greatful for the in-depth comments of the Reviewer 2, and have corrected the manuscript accordingly. All the suggested stilistic comments, as well as additional references have been added, enhancing the readability and specificity of the review.

Sincerely,

Juuli Raivola

Reviewer 3 Report

This review from one of the pioneering and leading labs in JAK biology is of potential high interest for a readership ranging from clinical and molecular oncologists to #aficionados# of JAK-STAT signaling. However, the review gives the impression that it is was largely compiled by an early-stage career scientist and will greatly improve by thorough revision:

  • removal of style/grammar/spelling/typo errors including usage of (in)definite articles, pronouns, 3rdperson #s#, english vs american etc.
  • guiding the reader through the topics by more clear distinction between ‘hallmark’ observations and otherwise interesting/complementing reports.

for time constraints, the reviewer mainly exemplifies the subfield of TYK2:

  • 2.1. Fusion proteins, paragraph starting at line 181: the authors should include the hallmark papers of Roberts et al 2014 (https://doi.org/10.1056/NEJMoa1403088) and Crescenzo et al 2015 (https://doi.org/10.1016/j.ccell.2015.03.006) describing the first patient-derived NGS-identified TYK2 fusions.
  • 2.2 Mutations in JAK…, paragraph starting at line 238: update the literature concerning TYK2 mutations causative for/associated with leukaemia e.g. Turrubiartes-Martinez et al 2020 (https://doi.org/10.3390/genes11121434)
  • state more precisely molecular axes or connections, e.g. line 247-8 description of Prutsch paper: correct description would be IL-10/22-TYK2-STAT1-BCL2 family member or MCL1 since BCL2 itself is grossly unaffected…
  • remove redundancies, e.g. 2.1. line 181-2 and 2.2 line 255-6, identical statement and wording of 2 sentences…
  • the readership coming from clinical and non-clinical background would appreciate information about the original specimen the mutated JAK effects were described in, i.e. primary patients samples or cultured cell lines

As stated above the reviewer only lists style and content deficiencies obvious by only focusing at specific parts of the review and suggests to check the other sections for similar drawbacks.

Author Response

We thank the Reviewer for the comments and have especially considered the TYK2 in section 2.1. and 2.2. As suggested, updated references were added, together with the hallmark references regarding TYK2 fusion proteins. In addition, corrections and modification have been done in the grammar more precise language is now used throughout the text. We believe these modifications have enhanced the review.

Sincerely,

Juuli